



# A High-Resolution Antarctic Grounding Zone Product from ICESat-2 Laser Altimetry

Tian Li[1], Geoffrey J. Dawson[1], Stephen J. Chuter[1], Jonathan L. Bamber[1]

[1]Bristol Glaciology Centre, School of Geographical Sciences, University of Bristol, Bristol, BS8 1SS, UK

*Correspondence to*: Tian Li (tian.li@bristol.ac.uk)

**Abstract.** The Antarctic grounding zone, which is the transition between the fully grounded ice sheet to freely floating ice shelf, plays a critical role in ice sheet instability, mass budget calculations and ice sheet model projections. It is therefore important to continuously monitor its location and migration over time. Here we present the first ICESat-2-derived high-resolution grounding zone product of the Antarctica Ice Sheet, including three important boundaries: the inland limit of tidal

flexure (Point F), inshore limit of hydrostatic equilibrium (Point H) and the break-in-slope (Point $I_b$). This dataset was derived from automated techniques developed in this study, using ICESat-2 laser altimetry repeat tracks between 30 March 2019 and 30 September 2020. The new grounding zone product has a near complete coverage of the Antarctica Ice Sheet with a total of 21346 Point F, 18149 Point H and 36765 Point $I_b$ identified, including the difficult to survey grounding zones, such as the fast-flowing glaciers draining into the Amundsen Sea Embayment. The locations of newly derived ICESat-2

landward limit of tidal flexure agree well with the most recent differential synthetic aperture radar interferometry (DInSAR) observations in 2018, with the mean absolute separation and standard deviation of 0.02 and 0.02 km, respectively. By comparing the ICESat-2-derived grounding zone with the previous grounding zone products, we find up-to 15 km grounding line retreat on the Crary Ice Rise of Ross Ice Shelf and the pervasive landward grounding line migration along the Amundsen Sea Embayment during the past two decades. We also identify the presence of ice plain on the Filchner-Ronne Ice Shelf and

the influence of oscillating ocean tides on the grounding zone migration. The product derived from this study is available at https://doi.org/10.5523/bris.bnqqyngt89eo26qk8keckglww (Li et al., 2021) and is archived and maintained at the National Snow and Ice Data Center.

## 1 Introduction

With a global sea level rise equivalent of 58 m (Fretwell et al., 2013), the Antarctica Ice Sheet has been losing ice at an

accelerated pace (Shepherd et al., 2018). This mass loss is largely driven by the ice dynamics of the marine ice sheet due to sustained and accelerated thinning of the ice shelves (Bamber et al., 2009; Paolo et al., 2015; Pattyn and Morlighem, 2020; Favier et al., 2014; Gardner et al., 2018) and rapid retreats of the grounding line (hereinafter referred to as the GL) (Point G in Fig. 1) (Christie et al., 2018; Milillo et al., 2019; Rignot et al., 2014; Scheuchl et al., 2016), which is the boundary between the grounded ice sheet and the floating ice shelves (Rignot et al., 2011a). The grounding line is identified as an





essential climate variable that is critical in understanding Earth's climate by the Global Climate Observing System (GCOS). Knowledge of its location is important in ice sheet numerical modelling and mass budget estimation as it controls the rates of ice flux from the grounded ice sheet into the ocean (Schoof, 2007) and it is a key indicator of the marine ice sheet instability (DeConto and Pollard, 2016; Joughin et al., 2014; Ritz et al., 2015). Therefore, continuous long-term monitoring of GL location and its temporal migration is crucial for understanding ice sheet stability and assessing Antarctica Ice Sheet's

contribution to future sea level rise.

The GL is located inside the grounding zone (hereinafter referred to as GZ). The GZ is defined as the region between the landward limit of tidal flexure (Point F in Fig. 1 of Brunt et al. (2010b)) where the ice is not influenced by ocean tides, and the inshore limit of hydrostatic equilibrium (Point H in Fig. 1 of Brunt et al. (2010b)) where the ice is floating in fully hydrostatic equilibrium (Brunt et al., 2010b; Fricker and Padman, 2006). Inside the GZ, there is often a surface elevation

minimum (Point $I_m$) and an inflection of ice surface slope where the slope changes most rapidly (Point $I_b$) (hereinafter referred to as the break-in-slope). As the GL is a subglacial feature, it is difficult to directly identify from in situ measurements or satellite observations (Horgan and Anandakrishnan, 2006). Instead, previous methods used satellite-observable grounding zone features (Points F and $I_b$) as proxies for the grounding line (Brunt et al., 2010b) (Fig. 1 in Brunt et al. (2010b)). Additionally, Point H is usually mapped as it can provide a measure of the grounding zone width and is

valuable in calculating ice thickness based on hydrostatic equilibrium (Dawson and Bamber, 2020; Rignot et al., 2011a).

There are two established approaches for estimating grounding line location using remote sensing techniques: a) directly detect the break-in-slope (hereinafter referred to as 'static method'); b) use observations of surface elevation change due to variations in ocean tide induced tidal flexure (hereinafter referred to as 'dynamic method'). The break-in-slope is mapped by identifying the inflection of the ice surface slope from a digital elevation model (DEM) (Brunt et al., 2010b, 2011; Fricker

and Padman, 2006; Horgan and Anandakrishnan, 2006) and the change in brightness on satellite optical imagery (Bindschadler et al., 2011; Christie et al., 2016, 2018). The satellite optical imagery based approaches are able to provide complete coverage of the Antarctica Ice Sheet (Bindschadler et al., 2011; Scambos et al., 2007). However, they work best only when the ice thickness increases rapidly inland from the grounding zone, and often fail to map the grounding line in areas of fast ice flow where the subglacial bed and surface slope are shallow (Christie et al., 2016, 2018).

The repeat track analysis and crossover analysis of satellite altimetry (Brunt et al., 2010b, 2011; Dawson and Bamber, 2017, 2020; Fricker and Padman, 2006; Li et al., 2020) and the differential synthetic aperture radar interferometry (DInSAR) (Brancato et al., 2020; Mohajerani et al., 2021; Rignot et al., 2016; Rignot, 1998; Scheuchl et al., 2016), use the dynamic method to detect Points F and H. Among which, DInSAR has been the most successful method of capturing Point F accurately and providing overall good spatial coverage. However, there are few regions have been measured repeatedly by

DInSAR (Friedl et al., 2020; Hogg et al., 2018), while some areas have not been mapped due to orbital limitations of the satellites (Mohajerani et al., 2018). Satellite altimetry, therefore, can provide valuable information where DInSAR measurements are not available. The existing satellite altimetry-derived grounding zone products from ICESat (Brunt et al.,



2010a) and CryoSat-2 (Dawson and Bamber, 2020) suffer from poor temporal and spatial coverages and are not suitable to monitor changes of the Antarctica grounding zone. The ICESat-2, launched on 15 September 2018, has higher along-track

resolution and better spatial coverage compared with ICESat (Markus et al., 2017). It can be used to map the Antarctica GZ with greater accuracy and spatio-temporal coverage than previous satellite altimetry-derived products. Here we generated the first ICESat-2-derived Antarctica grounding zone product with high spatio-temporal coverage using 18 months of ICESat-2 laser altimetry data (Li et al., 2021), including three grounding zone features Points F, H and $I_b$. This will be a valuable resource for comparison with other methods, and can provide high resolution GZ coverage in regions where DInSAR

measurements of Point F are not available (either spatially or in time). The new dataset also provides the state-of-the-art knowledge of GZ locations and is useful in understanding the Antarctica Ice Sheet instability.

This paper provides a detailed description of the ICESat-2-derived GZ product and the methodologies used to derive the dataset. We also discuss the associated uncertainties and validate the new grounding zone product with ICESat-2 crossover measurements and previous grounding zone products.

## 75   2 Data and methodology

### 2.1 ICESat-2 data and processing

ICESat-2 measures the ice sheet surface elevation at a repeat cycle of 91 days. The Advanced Topographic Laser Altimeter System (ATLAS) onboard ICESat-2 has three beam pairs in comparison with the single beam of the Geoscience Laser Altimeter System (GLAS) onboard ICESat. The across-track spacing between each beam pair is approximately 3.3 km with

a pair spacing of 90 m. The along track sampling interval of each beam is 0.7 m with a nominal 17 m diameter footprint (Markus et al., 2017; Smith et al., 2019).

In this study, we used version 3 of the ATL06 Land Ice Along-Track Height Product (Smith et al., 2019) from 30 March 2019 to 30 September 2020 (Scheick et al., 2019) to map three different GZ features, including the landward limit of tidal flexure (Point F), the inshore limit of hydrostatic equilibrium (Point H), and the break-in-slope (Point $I_b$) (Fig. 1 in Brunt et

al. (2010b)). The ATL06 elevation is calculated by averaging individual photon data over 40 m length segments with an along-track resolution of 20 m (Smith et al., 2019), the elevation accuracy is estimated to be better than 3 cm (Brunt et al., 2019). There are seven repeat cycles (3-9) in the study period, among which, cycles 4 and 9 are not complete.

We processed the ATL06 elevation data using the same methods described in Li et al. (2020). We did not apply the ocean tide correction to ICESat-2 ATL06 elevation and 're-tided' the ocean loading tide. Poor-quality elevation measurements

caused by clouds or background photon clustering were removed by applying the ATL06_quality_summary flag (Smith et al., 2019). A neighboring surface elevation consistency check was applied by using the along-track slope of each ground track. We only kept elevation measurements where differences between the original elevations and the estimated elevations



from along-track slope were lower than 2 m. The reference segment locations of each ground track were also derived from the 'segment_quality' group to calculate a reference track, which will later be used in the grounding zone calculation.

## 2.2 Repeat track preparation

Our method of estimating GZ features utilizes ICESat-2 repeat tracks from different cycles (Fig. 1 Box 1). Following the steps of repeat track generation described in Li et al. (2020), the surface elevation, elevation measurement geolocations, and the reference segment geolocations of six ground tracks along each of the 1387 Reference Ground Tracks (RGTs) were categorized into nine distinct repeat-track data groups, including six single-beam repeat-track data groups and three beam-pair repeat-track data groups (Figs. 4a and 4b in Li et al. (2020)). For each repeat-track data group, a 'nominal reference track' was calculated by averaging the locations of reference segments from all repeat tracks inside this data group. A reference GL was also calculated as the intersection between the nominal reference track and a composite GL which was generated by merging the Depoorter et al. (2013) GL with the most recent grounding lines from different sources (Table A1). Allowing for a possible GL change between the current GZ location and the composite GL, we defined a 15 km calculation window landward and seaward of the reference GL along the nominal reference track, only ATL06 elevation measurements located within this calculation window were used in the GZ calculation (Li et al., 2020). This is to ensure the pre-defined calculation window can capture the grounding zone adequately in our study period due to potential grounding line changes during the past decade, especially the fast-flowing glaciers.

We removed ATL06 data points with elevation higher than 400 m and data points located in open water based on the coastline mask provided in the SCAR Antarctic Digital Database (ADD) (https://data.bas.ac.uk/items/ed0a7b70-5adc-4c1e-8d8a-0bb5ee659d18/, last access: 6 July 2020) to only include data in GZ. We also only included repeat tracks inside the calculation window that contain at least 50% valid elevation measurements, as any track with less than 50% data was regarded as insufficient for GZ calculation and was removed.

**Figure 1: The automatic workflow of identifying the grounding zone features from ICESat-2 data. (Box 1) ICESat-2 repeat track preparation; (Boxes 2 and 3) Estimation of the landward limit of tidal flexure (Point F) and the inshore limit of hydrostatic equilibrium (Point H) from the dynamic method; (Boxes 4 and 5) Estimation of the break-in-slope Point $I_b$ from the static method. (Box 6) ICESat-2 crossover analysis. Grey boxes denote the grounding zone features. Boxes with other colors denote key steps in the grounding zone estimation.**

## 2.3 Dynamic method: identify the limits of tidal flexure

The key feature of dynamic method is to identify the temporal changes in ice surface elevation due to ocean tides between Points F and H from different repeat tracks (Brunt et al., 2010b, 2011; Fricker and Padman, 2006). The temporal ice surface elevation changes were derived from a set of 'elevation anomalies' (Fig.1 Box 2). For each single-beam repeat-track data group, the reference elevation profile along the nominal reference track was first calculated by averaging the elevations of each repeat track at the nominal reference track, then elevation anomalies were calculated by differencing the elevation profile of each individual repeat track and this reference elevation profile (Li et al., 2020) (Figs. 2c, 2h, 2m). For the beam-





pair repeat-track data group, the elevation profile of each individual repeat track was first corrected for the across-track slope onto the nominal reference track (Eq. (1) and (2) in Li et al. (2020)). The average of all across-track slope corrected elevations from each track at the nominal reference track was then taken as the reference elevation profile. The elevation

anomalies were calculated by subtracting this reference elevation profile from the across-track slope corrected elevation profile of each repeat track inside the beam-pair repeat-track data group.

The estimation of GZ features Points F and H are based on extracting the transition points from the mean absolute elevation anomaly (MAEA), which is defined as the average of the absolute value of all elevation anomaly profiles (Figs. 2d, 2i, 2n). The inland limit of tidal flexure, Point F, is identified as the point where the elevation anomaly of each repeat track exceeds a

noise threshold (Brunt et al., 2010b, 2011; Fricker et al., 2009). The region where the MAEA is close to zero is regarded as the fully grounded ice (the region to the left of Point F in Fig. 1 of Brunt et al. (2010b)), as it is not influenced by tidal motion. Point F was then estimated to be the point where the gradient of the MAEA first increases from zero and the second derivative of the MAEA reaches its positive peak (Li et al., 2020). The inshore limit of hydrostatic equilibrium, Point H, is identified as the location where the elevation anomaly of each repeat track reaches its maximum and becomes stable. It was

estimated as the transition point where the gradient of the MAEA finally decreases to zero and the second derivative of the MAEA reaches its negative peak (Li et al., 2020).

To select the correct transition points from the second derivative of the MAEA curve as Points F and H, previously we used an error function fit to the MAEA as a guide (Li et al., 2020). While the error function can reliably estimate Point H because the gradient of elevation anomaly at which always changes smoothly to zero, it is unreliable in identifying Point F where

there is a sharp transition on the MAEA curve or the across-track slope related noises on land ice is high (Green dots in Figs. 2j and 2o). To solve the inaccurate picks of Point F under these circumstances, instead of using error function fitting, we used a three-segment piecewise function fitting only to the landward part of the Point H on the MAEA profile (Fig. 1 Box 3) (Green lines in Figs. 2g, 2j, 2o). The closest positive peak of the second derivative of this piecewise function to the reference GL was taken as a guide point to find Point F. As a final step, all results are visually inspected due to the complex nature of

the grounding zones, and the ICESat-2 crossover measurements are used as a reference at Filchner-Ronne Ice Shelf and Ross Ice Shelf (Section 2.5). In the final GZ product, we also recorded the number of repeat cycles used and the ocean tide range calculated as the maximum elevation anomaly deviation from all repeat tracks at Point H.

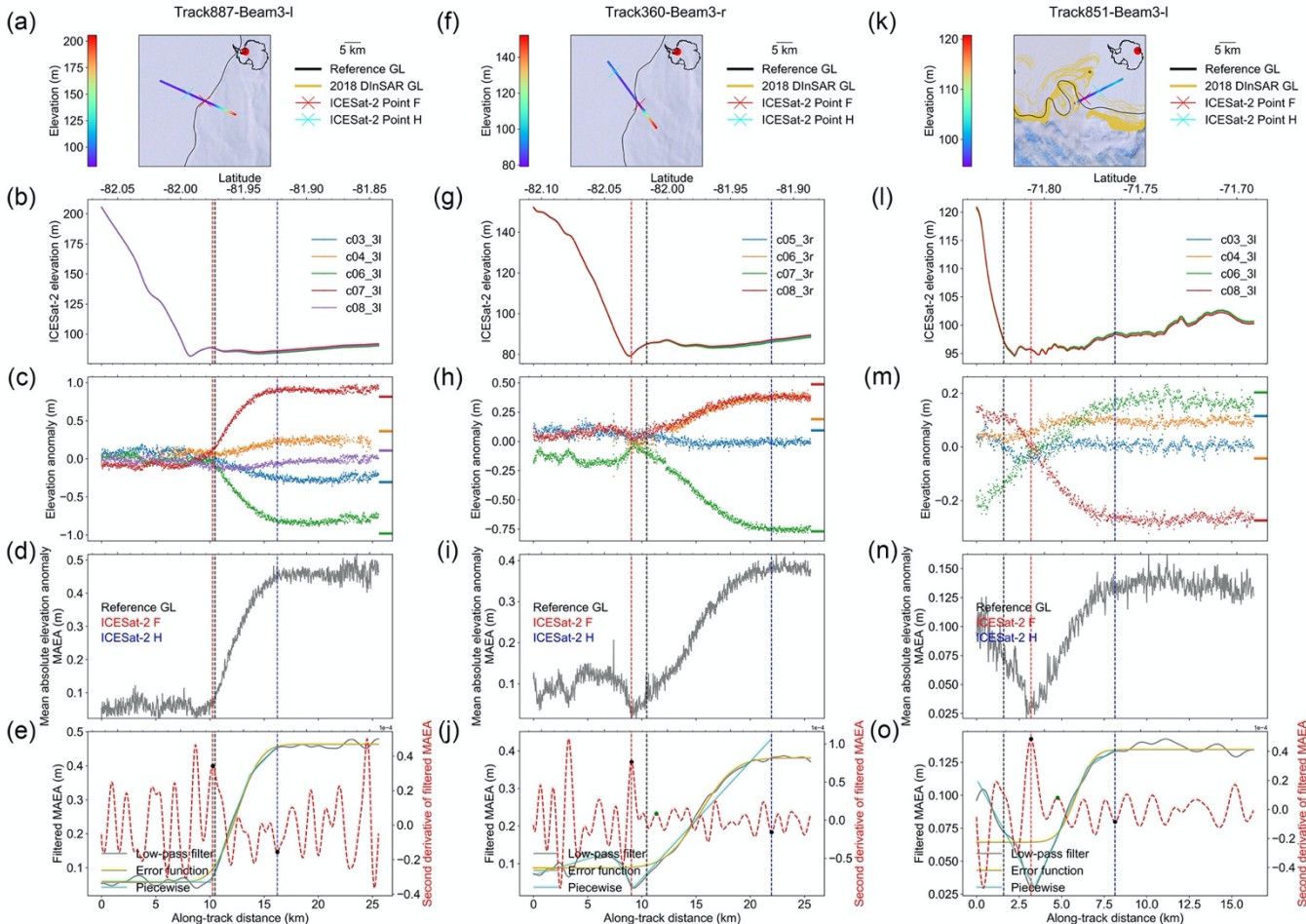

**Figure 2: Examples of repeat track analysis for three tracks: track 887 (a-e) and track 360 (f-j) on the Filchner-Ronne Ice Shelf,**
**track 851 (k-o) on the Amery Ice Shelf. (a, f, k) The locations of ICESat-2-derived inland limit of tidal flexure Points F (red cross)**
**and the inland limit of hydrostatic equilibrium H (cyan cross), along with the reference elevation of the nominal reference track**
**(color-coded), the composite grounding line (GL) (black line) used to calculate the reference GL in the repeat track analysis, the**
**DInSAR-derived Point F in 2018 (yellow line) (Mohajerani et al., 2021), all data are overlaid on the Landsat Image Mosaic of**
**Antarctica (Bindschadler et al., 2008). (b, g, l) ICESat-2 're-tided' elevation profiles. (c, h, m) The elevation anomalies of each**
**repeat track inside each repeat-track data group, the horizontal lines at the right are the zero mean tide height predictions from**
**the CATS2008 tidal model (Padman et al., 2002). (d, I, n) The mean absolute elevation anomaly (MAEA). (e, j, o) Low-pass filtered**
**MAEA is shown as a grey solid line, error function fitting of the MAEA is shown as a yellow solid line, the three-segment piecewise**
**function fitting of the landward part of the Point H of the MAEA is shown as a green solid line, the second derivate of low-pass**
**filtered MAEA is shown as a red dashed curve, the piecewise function derived Point F is shown as the black dot on the right, the**
**wrong Point F picks from error function fitting are shown as the green dots in (j) and (o), the Point H is shown as the black dot on**
**the left of each panel. Locations of Point F, Point H and the reference GL are marked as the vertical dashed red line, vertical**
**dashed blue line, and vertical dashed black line in all panels apart from (a, f, k).**



## 2.4 Static method: identify the break-in-slope

The break-in-slope Point $I_b$ and elevation minimum Point $I_m$ are the points where the slope changes most rapidly and where
the slope is zero inside the grounding zone (Bindschadler et al., 2011), respectively. Previous studies using ICESat laser altimetry data selected the break-in-slope by hand (Brunt et al., 2010b; Fricker and Padman, 2006), however given the increased data volume available for ICESat-2, the manual approach is no longer feasible. Here we developed an automated technique to select the break-in-slope (Fig. 1 Boxes 4 and 5) by solving complex surface morphologies of the grounding zone, such as crevasses and ice plain which used to impose difficulty on interpreting break-in-slope (Brunt et al., 2010b,
2011; Fricker et al., 2009; Horgan and Anandakrishnan, 2006). We only used the single-beam repeat-track data group to determine the break-in-slope since we do not need to calculate elevation changes between repeat tracks, which can be influenced by the across-track slope induced errors.

The break-in-slope Point $I_b$ is often associated with a local topographic minimum Point $I_m$ inside the GZ (Brunt et al., 2010b). According to this, we first estimated the potential Point $I_m$ using the along-track RMS height R, then used this
potential Point $I_m$ to derive the break-in-slope Point $I_b$. The RMS height of the along-track topography, also referred to as the standard deviation of the elevation, has proved to be a robust way of estimating the surface roughness at finer scale (Cooper et al., 2019). It is more sensitive to identifying the local topographic extremes, compared with using the reference elevation profile itself.

After obtaining the reference elevation profile on the nominal reference track of each single-beam repeat-track data group
(Figs. 3b, 3h, 3n), we first linearly interpolated the reference elevation based on sequential segment id at an along-track distance of 20 m to fill the data gaps. To remove noise caused by small-scale topographic features such as crevasses, we applied a Butterworth low-pass filter with normalized cut-off frequency of 0.032 and an order of 5 to the interpolated reference elevation profile (black lines in Figs. 3c, 3i, 3o). The low pass filter only removed the high-frequency noise without changing the shape of the reference elevation profile, thus it will not alter the locations of grounding zone features.

We calculated the along-track RMS height R with a bin size of 100 m (5 elevation measurements) of the low-pass filtered reference elevation profile using Eq. 1 (Cooper et al., 2019),

$$R = \left[ \frac{1}{n-1} \sum_{i=1}^{n} (z(x_i) - \bar{z})^2 \right]^{\frac{1}{2}}, \tag{1}$$

where $n$ is the number of sample elevation points, $z(x_i)$ is the elevation of each point, $\bar{z}$ is the average elevation of all data points in the calculation window. R is given for the mid-point of each calculation window. The examples of the along-track
RMS height for three different tracks located in Amundsen Sea Embayment are shown as black line in Figs. 3d, 3j and 3p. The negative peaks of RMS height with value less than 0.5 m were taken as local topographic extremes. They were further filtered to only keep the elevation minimums based on the elevation peaks of the reference elevation profile. To find the potential Point $I_m$, we first fitted a four-segment piecewise function to the reference elevation profile (yellow line in Figs. 3c,



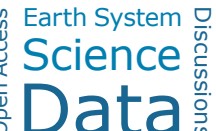

3i and 3o). The closest positive peak of its second derivative to the reference GL was taken as a guide point to find the
potential Point $I_m$ from local elevation minimums.

The along-track surface slope (Figs. 3e, 3k and 3q), and the slope breaks (Figs. 3f, 3l and 3r) which are the first gradients of
the along-track slope, were calculated from the low-pass filtered reference elevation profile. A group of peaks were
identified from the absolute values of the slope breaks as potential break-in-slope features (red crosses in Figs. 3f, 3l and 3r),
as they are the locations where the along-track slopes change most rapidly. The break-in-slope Point $I_b$ (black dots in Figs.
3f, 3l and 3r) was then taken as the highest slope break between the two closest slope breaks to the potential Point $I_m$
identified at the previous step. We visually checked all the break-in-slope estimations as a final step. The complete algorithm
workflow was demonstrated over three typical regions: slow moving region with steep slope (track 1377 on the western
flank of Bear Peninsula, Figs. 3a-f), highly-crevassed fast-flowing glacier (track 515 on the 'butterfly' region of Thwaites
Glacier, Figs. 3g-l) and the ice plain of fast-flowing glacier (track 211 on an unnamed fast-flowing glacier at Getz Ice Shelf,
Figs. 3m-r), proving our method can reliably detect the break-in-slope of grounding zones with different surface
morphology.



**Figure 3: Estimation of break-in-slope (Point I_b) from ICESat-2 repeat tracks on the Amundsen Sea Embayment. (a - f) track 1377 on Bear Peninsula; (g - l) track 515 on the 'butterfly' region of Thwaites Glacier; (m - r) track 211 on an unnamed glacier of Getz**
**Ice Shelf. Geolocations and elevations of ICESat-2 elevation profiles for three tracks are shown in (a, g and m) superimposed over the on the Landsat Image Mosaic of Antarctica (Bindschadler et al., 2008), the reference GL is shown as black line, the ASAID break-in-slope is shown as the green line (Bindschadler et al., 2011), the Sentinel-1a/b DInSAR-derived grounding line is shown as the yellow line (Mohajerani et al., 2021), the ICESat-2-derived break-in-slope is shown as red cross. (b, h, n) The reference elevation profile along the nominal reference track. (c, I, o) The interpolated reference elevation profile is shown as the solid cyan**
**line, the low-pass filtered interpolated reference elevation profile is shown as the solid black line, the four-segment piecewise function fitting is shown as the solid yellow line. (d, j, p) The along-track RMS height of the reference elevation profile, regional elevation minimums are shown as the blue crosses. (e, k, q) The along-track slope of reference elevation profile. (f, l, r) The absolute slope break along the reference elevation profile, peaks of the slope break are shown as red crosses, the final break-in-slope Point I_b is shown as the black dot. Locations of Point I_b and the reference GL are marked as the vertical dashed red line and**
**the vertical dashed black line in all panels apart from (a, g, m).**



## 2.5 Crossover analysis

To validate the repeat-track analysis derived GZ features, we calculated the elevation changes at crossovers from ICESat-2 ascending and descending tracks (Fig.1 Box 6). This can be used to measure the grounding line (Li et al., 2020), which is the boundary between high elevation changes on floating ice due to tidal movement and low elevation changes on land ice not influenced by ocean tides. In this study, the crossover analysis was performed at the two largest ice shelves in Antarctica with the highest crossover densities, the Filchner-Ronne Ice Shelf and the Ross Ice Shelf. To calculate the elevation changes at crossovers, we closely follow the methodology developed in Li et al. (2020). When removing the crossovers with time stamps of the ascending and descending track on floating ice in the same tidal phase, we set the minimum threshold of elevation change due to ocean tides on floating ice to be 20 cm, as the minimum detectable tidal amplitude from repeat-track analysis is around 10 cm in Filchner-Ronne Ice Shelf and Ross Ice Shelf. After deriving the mean elevation difference at each crossover, we interpolated them onto a 2 km regular polar stereographic grid using a distance-weighted gaussian kernel. The correlation length of the gaussian kernel is 5 km and it uses the nearest 100 measurements. For the final gridded crossover elevation changes, we set a threshold of 20 cm for the location where the ice starts to be affected by ocean tides, which is the Point F. We are aware that the elevation change threshold of Point F is not constant across all the regions of these two ice shelves, however the 20 cm threshold represents the most conservative estimation of Point F location, that is the crossover grid with an elevation change value less than 20 cm should be grounded ice.

## 2.6 Uncertainty assessment

The highest absolute precision in identifying the grounding zone features Points F, H and $I_b$ from ICESat-2 repeat tracks is constrained by the 20 m along-track separation along each beam. However, the measurement error varies with different track as the geophysical conditions are different, and several factors need to be considered when evaluating the uncertainty of each GZ feature identified using the techniques developed in this study. (1) The selection of specific repeat tracks used in the GZ calculation will result in different tidal amplitudes, low tidal amplitude will decrease signal-to-noise ratio of the elevation anomaly and thus influence the estimation of Points F and H. (2) The across-track slope induced elevation change will be high in some high relief regions, although the typical across-track separation of ICESat-2 repeat tracks is approximately 10 m in Antarctica (Li et al., 2020) and an across-track slope correction is applied at the nominal reference track. (3) If melt ponds exist, ATL06 will normally identify the flat water surface instead of the underlying ice surface (Fricker et al., 2020). This will result in high elevation anomaly due to changes in melt pond surfaces across different melt seasons captured by different repeat cycles. (4) Orientation of the repeat tracks relative to the grounding zone. (5) Ice surface roughness such as crevasses and rifts can introduce noises in the elevation anomaly profiles (Brunt et al., 2010b), compromise the ability to identify limits of tidal flexure inside the grounding zone. Besides, the high slopes inside the crevasses and rifts can contaminate the break-in-slope signal (Horgan and Anandakrishnan, 2006). (6) Ice surface feature advection across the grounding zone due to high ice velocity will also introduce noises in elevation anomaly (Fricker et al., 2009).



To estimate the positional uncertainty of the grounding zone features, we compare the results calculated along the left and right beams, as well as the nominal reference track in each beam pair. As the left and right beams are only separated by

approximately 90 m and the GZ identified from the repeat-track analysis for beam pair often locates in the middle between the left and right beams (~45 m in either direction), we do not expect here exist large deviations between these three GZs (Li et al., 2020). The standard deviation between the locations of Point F at the left and right beams for the whole Antarctica Ice Sheet is 66.27 m, while the standard deviation of Point F between the single beam and the nominal reference track in the middle of the beam pair is 84.67 m (Table 1). For Point H, the standard deviations for these two comparisons are 519.12 m

and 560.59 m, respectively (Table 1). Since the static method of calculating the break-in-slope does not use the beam-pair repeat-track data group, we calculated the separations of the break-in-slope derived along the left beam and the right beam inside the same beam pair, and the standard deviation for Point $I_b$ is 12.3 m (Table 1). Thus, we assign the typical uncertainties for the ICESat-2-derived Points F, H and $I_b$ to be 80 m, 560 m and 10 m.

**Table 1: Mean absolute separations and standard deviations between the grounding zone features calculated from the single-beam**
**repeat-track data group and beam pair repeat track data group.**

|  | Point F | | Point H | | Point $I_b$ | |
|---|---|---|---|---|---|---|
|  | Mean absolute separation (m) | Standard deviation (m) | Mean absolute separation (m) | Standard deviation (m) | Mean absolute separation (m) | Standard deviation (m) |
| Left beam vs. right beam | 153.69 | 66.27 | 583.41 | 519.12 | 102.9 | 12.3 |
| Single beam vs. beam pair | 121.16 | 84.67 | 499.64 | 560.59 | - | - |

## 3 Results

### 3.1 Antarctica grounding zone distributions

Using the grounding zone mapping techniques developed in this study, we produced a new high-resolution grounding zone product (Li et al., 2021) by identifying 21346 Point F (Fig. 4a), 18149 Point H (Fig. 4b) and 36765 Point $I_b$ (Fig. 4c) over the

Antarctica Ice Sheet from 18 months of ICESat-2 repeat tracks. The dataset is comprised of three CSV files, one for each grounding zone feature. Every file contains columns 'lat', 'lon', 'track', 'beam_pair', 'beam', 'repeat_cycle_no' to denote the latitude and longitude of the grounding zone feature, the RGT number, beam pair number, beam number and the number of repeat cycles used in the grounding zone calculation. For Points F and H, they contain an additional column 'tide_range' which is the tidal range derived at the Point H from elevation anomalies.

Compared with the ICESat-derived grounding zone product (Brunt et al., 2010a), which has 1497 Point F, 1470 Point H and 1493 Point $I_b$, the ICESat-2-dervied grounding zone features in this study have greatly improved the GZ density and coverage, including the previously poorly mapped regions such as fast flowing ice streams in Amundsen Sea Embayment.





For Points F and H, we obtained near complete coverages on the Larsen C Ice Shelf, Filchner-Ronne Ice Shelf, Dronning Maud Land, Ross Ice Shelf, Sulzberger Ice Shelf, including numerous ice rises and ice rumples (Figs. 4a and 4b). Compared
with Points F and H, the ICESat-2-derived Point $I_b$ further improves the GZ coverage. It is able to recover the grounding zone of the fast flowing glaciers that are difficult to map with the dynamic method, including the Pine Island Glacier, Thwaites Glacier, Kohler, Smith and Pope Glaciers located in the Amundsen Sea Embayment, as well as the mountainous regions at the Victoria Land (Fig. 4c). In addition, the ICESat-2-derived Point $I_b$ also provides complete coverage for the ice rises and ice rumples across the Antarctic Ice Shelves, which are not available from the ASAID product (Bindschadler et al.,
290 2011).

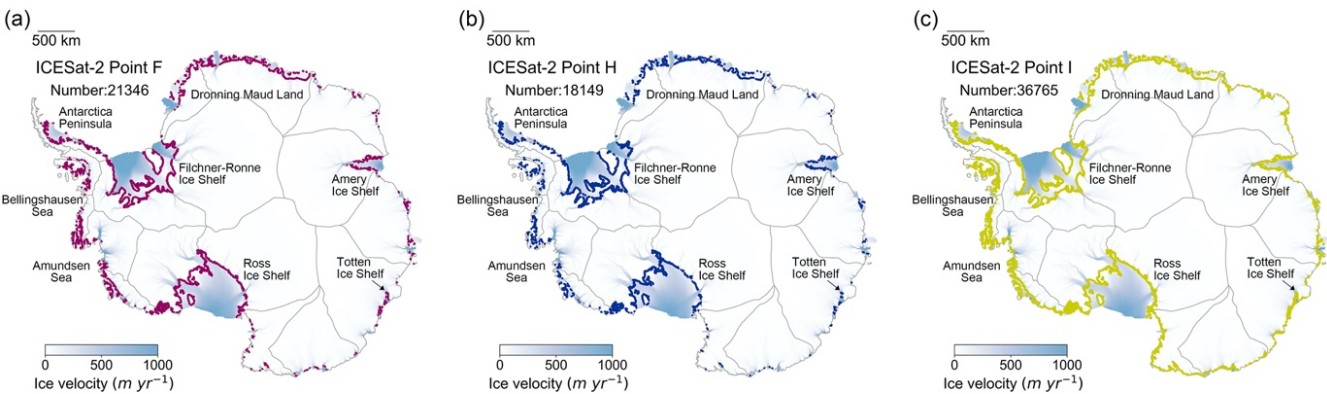

**Figure 4: Spatial distributions of ICESat-2-derived grounding zone features of the Antarctic Ice Sheet. (a) ICESat-2-derived inland limits of tidal flexure (Point F; purple dots). (b) ICESat-2-derived inland limit of hydrostatic equilibrium (Point H; blue dots). (c) ICESat-2-derived break-in-slope (Point $I_b$; yellow dots). In all subplots, data are superimposed over recent ice velocity**
**magnitudes (Rignot et al., 2017) and IMBIE basin boundary (Shepherd et al., 2018; Rignot et al., 2011b).**

**3.2 Comparison with ICESat-2 crossover measurements**

The elevation changes at the crossovers in Filchner-Ronne Ice Shelf and Ross Ice Shelf were mapped in our study (Figs. 5, 6, A1 and A2). The transitions from land ice (low |dh|) to floating ice (high |dh|) at the crossovers can show the approximate location of the grounding line (Li et al., 2020), with which we compared our repeat-track-derived GZ results. In general, the
crossover-derived grounding line, where the |dh| is 20 cm which is the minimum detectable tidal range in these two regions, show good agreement with the ICESat-2-derived Point F and $I_b$ (Figs. 5, 6, A1 and A2). At Hercules Inlet of the Filchner-Ronne Ice Shelf, the ICESat-2-derived Point F significantly improved the GZ coverage compared with DInSAR measurements (ESA, 2017; Rignot et al., 2016) and the ICESat-derived Point F (Fig. 5b). The ICESat-2-derived Point $I_b$ are able to image the complex inlets at the eastern flank where the ASAID Point $I_b$ failed to do so (Fig. A1b). On
Bungenstockrücken, there exists an approximate 3 km deviation between the ICESat-2-derived Point $I_b$ and the crossover-derived grounding line (Fig. A1c). This directly confirms the existence of an ice plain, which is defined as grounded ice with low surface slope adjacent to the GL and where the Point $I_b$ is several kilometers landward of Point F (Brunt et al., 2011).





Inside the ice plain, the ICESat-2-derived Point F show ephemeral grounding (Fig. 5c) which is likely to be caused by tidal variations (Brunt et al., 2011). On the main glacier trunk of the Support Force Glacier (Fig. 5d), the crossover-derived GL

and ICESat-2-derived Point F align well with the ESA CCI DInSAR-mapped Point F in 2016 and the CryoSat-2-derived Point F in 2017. On the western side of the glacier, the ICESat-2-derived Point F, crossover-derived GL, as well as the CryoSat-2-derived Point F, show an approximately 10 km seaward migration compared with the ESA CCI DInSAR-derived Point F in 2016. On the Foundation Ice Stream, the ICESat-2-derived Points $I_b$ align well with the crossover-derived GL, but locate about 20 km landward compared with ASAID Point $I_b$ (Fig. A1d). It is possible that the satellite imagery has difficulty

in mapping the break-in-slope of fast flowing ice streams where the surface slope change is less prominent. On the main glacier trunk of Bailey Ice Stream, the crossover-derived GL, ICESat-2-derived Point F, CryoSat-2-derived Point F, and the ESA CCI DInSAR-derived Point F in 2014, agree well with each other (Fig. 5e). However, on the western side of the Parry Peninsula (Fig. 5e), the ESA CCI DInSAR-derived Point F shows an approximately 10 km retreat compared with all the other GL measurements. In the same region, the ICESat-2-derived Points $I_b$ agree well with the crossover-derived GL

distribution, but the ASIAD Point $I_b$ shows an approximate 10 km retreat.

On the Ross Ice Shelf, the ICESat-2-derived Point F significantly improves the grounding line coverage compared with the MEaSUREs DInSAR measurements (Rignot et al., 2016) (Fig. 6a). On the Mercer Ice Stream (Fig. 6b), the ICESat-2-derived Point F, the crossover-derived GL, MEaSUREs and ESA CCI DInSAR measurements, as well as the CryoSat-2-derived Point F, agree well with each other. For Point $I_b$, the ICESat-2-dervied Point $I_b$ can capture the small inlets in this

region compared with the ASAID product (Fig. A2b). On Crary Ice Rise, both ICESat-derived Point F and Point $I_b$ agree well with the crossover GL distribution (Figs. 6c and A2c), but show an up-to 15 km retreat in this region compared with all the previous GZ measurements. On Siple Dome, the ICESat-2-dervied Point F and crossover-derived GL have good agreement with the previous GL products (Fig. 6d). However, the ICESat-2-derived Points $I_b$ extend seaward about 50 km compared with the ASAID-derived Point $I_b$ (Fig. A2d). On Echelmeyer Ice Stream, where there is only one ICESat-derived

Point F existed, the ICESat-2-derived Points F show an approximate 30 km retreat compared with ICESat-derived Point F, but agree well with the CryoSat-2-derived Point F in 2017 and the ICESat-2 crossover-derived GL (Fig. 6e), it further confirms the conclusion that the ICESat picked the wrong Point F in this region (Dawson and Bamber, 2017).

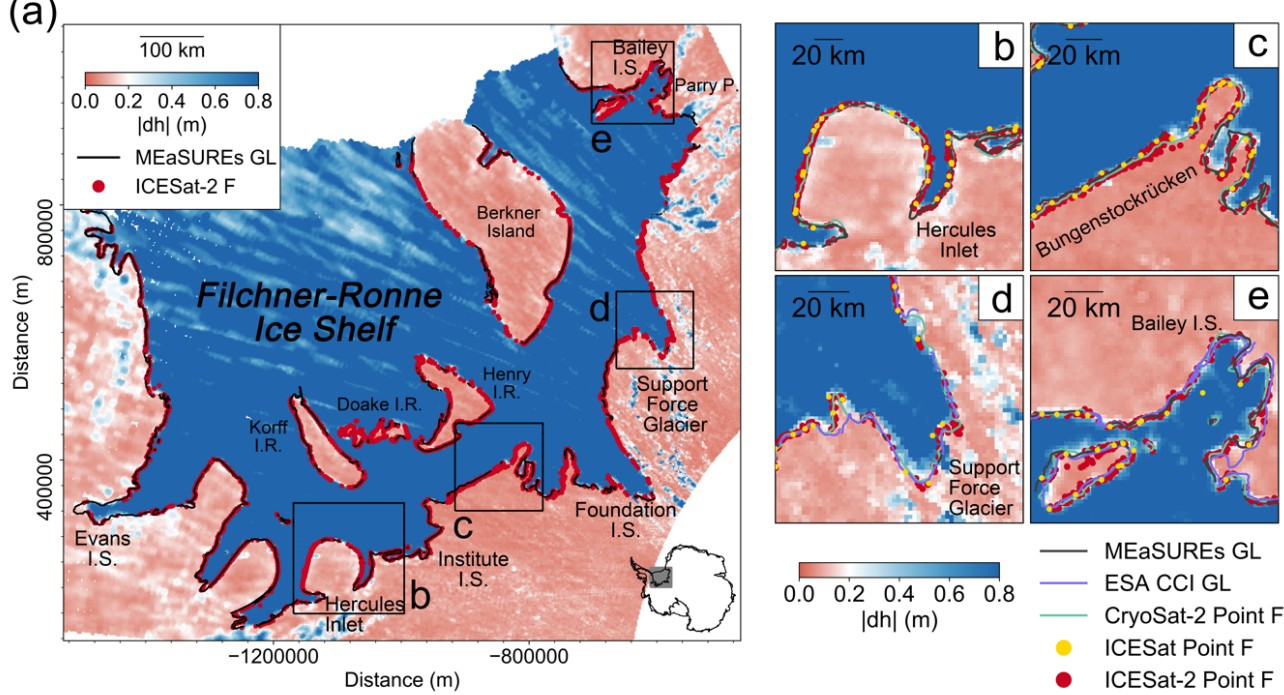

**Figure 5: (a)** Spatial distribution of the absolute elevation change at ICESat-2 crossovers per 2 km grid cell across the Filchner-Ronne Ice shelf, the four black boxes denote the individual regions plotted in b-e). b) Hercules Inlet; c) Bungenstockrücken; d) Support Force Glacier; e) Bailey Ice Stream. In all subplots, the ICESat-2-derived inland limits of tidal flexure (Point F) are shown as red dots. The ICESat-derived Points F are shown as the yellow dots. The MEaSUREs DInSAR-derived Point F (Rignot et al., 2016) is shown as the black line. The ESA CCI DInSAR-derived Point F is shown as the purple line. The CryoSat-2-derived Point F is shown as the green line (Dawson and Bamber, 2020).





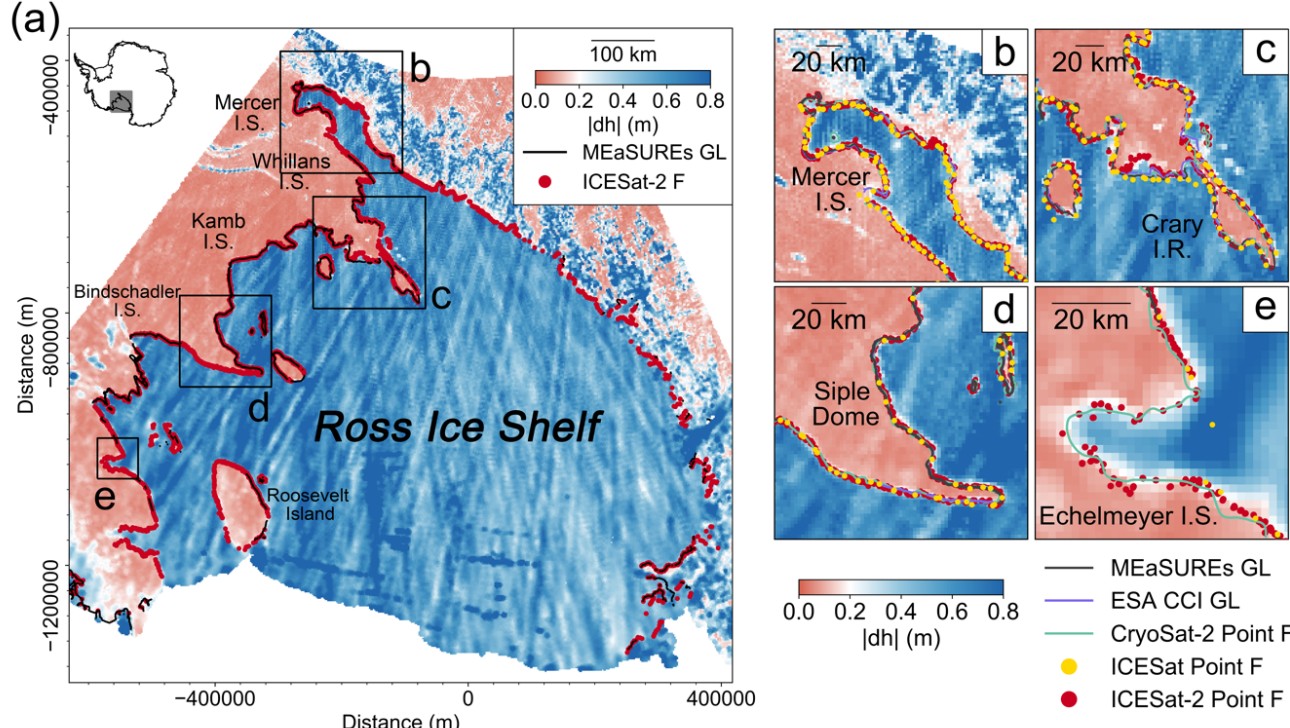


**Figure 6: (a) Spatial distribution of the absolute elevation change at ICESat-2 crossovers per 2 km grid cell across the Ross Ice shelf, the four black boxes denote the individual regions plotted in b-e). b) Mercer Ice Stream; c) Crary Ice Rise; d) Siple Dome; e) Echelmeyer Ice Stream. In all subplots, the ICESat-2-derived inland limits of tidal flexure (Point F) are shown as red dots. The ICESat-derived Points F are shown as the yellow dots. The MEaSUREs DInSAR-derived Point F (Rignot et al., 2016) is shown as** 345 **the black line. The ESA CCI DInSAR-derived Point F is shown as the purple line. The CryoSat-2-derived Point F is shown as the green line (Dawson and Bamber, 2020).**

## 3.3 Comparison with other grounding zone products

### 3.3.1 Inland limit of tidal flexure - Point F

We first compared the ICESat-2-derived Point F to the deep-learning-based grounding line product from Sentinel-1a/b DInSAR measurements in 2018 (Mohajerani et al., 2021). This is the latest pan-Antarctica grounding line product and is close in acquisition time to ICESat-2 (up to 1 year apart). Thus we do not expect large separations in GL locations between these two products, due to any changes in grounding line. The dataset has a precision of 200 m (Mohajerani et al., 2021), however due to limitation of Sentinel's coverage in polar regions, this product does not fully cover Filchner-Ronne Ice Shelf

and Ross Ice Shelf. The absolute separations between 2018 DInSAR-derived Point F with ICESat-2-derived Point F are shown in Fig. 7a. Despite the relatively small difference in measurement time, there may still be changes in Point F. In general, the rapid retreat of grounding line happens in fast ice flow (Konrad et al., 2018). Therefore, we also divided the



grounding line separations into two categories: slow moving regions where the ice velocity is less than 100 m yr$^{-1}$ (Fig. 7b) and fast flowing region where the ice velocity is higher than 100 m yr$^{-1}$ (Fig. 7c).

In total, the mean absolute separation and standard deviation across the ice sheet between the two products are 0.02 and 0.02 km, respectively, comparable to the precision of the DInSAR GL product (Table 2). This indicates that the ICESat-2-derived Point F can achieve the same level of precision compared with the DInSAR measurements. 84 % of the surveyed GZ locate in slow moving regions. As expected, the overall mean separations and standard deviations of slow moving regions where the grounding line is normally stable, are lower than in fast flowing regions. The increase in GL separation in fast flowing
regions between the two products is possibly due to the reduced ICESat-2 GL measurements caused by low signal-to-noise ratio in elevation anomalies of repeat tracks, and the fact that DInSAR often suffers from poor signal coherence due to high ice velocity. On Amundsen Sea Embayment and Bellingshausen Sea Sector, which have been experiencing severe mass loss and rapid grounding line retreat during past two decades (Bamber and Dawson, 2020; Milillo et al., 2017, 2019; Rignot et al., 2014, 2019; Scheuchl et al., 2016), the mean absolute separations in fast-flowing region are 0.17 km and 0.24 km,
respectively. The highest mean absolute separation and standard deviation, however, locate in Wilkes Land, East Antarctica (Table 2, Fig. 7a). The Moscow University Ice Shelf and Totten Glacier Ice Shelf in Wilkes Land are both narrow embayment with fast ice flow, where the ice amplitudes may not be in fully hydrostatic equilibrium and the high ice velocity can often lead to DInSAR measurement errors.

In slow moving regions, we observed large deviations between the two products such as the Dronning Maud Land (Figs. 7b
and 8a). They are possibly caused by the ephemeral grounding of ice on the scale of kilometers across the ice plain with low surface slope as the ocean tide rises and falls (Bindschadler et al., 2011; Brunt et al., 2011; Milillo et al., 2017). Here we took two examples to demonstrate the short-term grounding zone feature migrations induced by ocean tide oscillation. On the Novyy Island of Dronning Maud Land, the distance between the ICESat-2-derived Point F along the right beam of track 145 is about 2 km compared with the 2018 DInSAR-derived Point F (Mohajerani et al., 2021) (Figs. 8a), while the ICESat-2-
derived Point F along left beam of track 153 in the same region is only less than 100 m away from the DInSAR-derived Point F (Fig. 8f). The large difference in Point F location is not caused by the errors in methodology but due to the tidal variations on a lightly grounded ice plain in this region. The tidal range at Point F along track 145 is 0.41 m while it is 1.03 m at Point F along track 153. The observation suggests that the ice shelf is grounded at low tide and floating at high tide (Brunt et al., 2011).

**Table 2: Mean absolute separation (km) and standard deviation (km) between ICESat-2-derived landward limit of tidal flexure (Point F) and 2018 DInSAR-derived Point F (Mohajerani et al., 2021) in individual regions.**

| Region | Ice velocity < 100 m/yr | | | Ice velocity > 100 m/yr | | | All | |
|---|---|---|---|---|---|---|---|---|
| | Mean Absolute separation (km) | Standard deviation (km) | Ratio | Mean Absolute separation (km) | Standard deviation (km) | Ratio | Mean Absolute separation (km) | Standard deviation (km) |
| Antarctica | 0.02 | 0.02 | 0.84 | 0.09 | 0.1 | 0.16 | 0.02 | 0.02 |




| | | | | | | | | |
|---|---|---|---|---|---|---|---|---|
| Larsen C Ice Shelf | 0.02 | 0.02 | 0.94 | 0.13 | 0.14 | 0.06 | 0.02 | 0.02 |
| Dronning Maud Land | 0.07 | 0.08 | 0.84 | 0.11 | 0.14 | 0.16 | 0.07 | 0.08 |
| Amery Ice Shelf | 0.05 | 0.07 | 0.91 | 0.16 | 0.23 | 0.09 | 0.02 | 0.02 |
| Amundsen Sea | 0.01 | 0.01 | 0.6 | 0.17 | 0.21 | 0.4 | 0.03 | 0.03 |
| Bellingshausen Sea | 0.06 | 0.07 | 0.85 | 0.24 | 0.33 | 0.15 | 0.06 | 0.07 |
| Wilkes Land | 0.09 | 0.09 | 0.47 | 0.95 | 0.94 | 0.53 | 0.64 | 0.84 |
| Sulzberger Ice Shelf | 0.01 | 0.01 | 0.97 | 0.53 | 0.87 | 0.03 | 0.01 | 0.01 |
| George VI Ice Shelf | 0.01 | 0.01 | 0.8 | 0.12 | 0.14 | 0.2 | 0.01 | 0.01 |

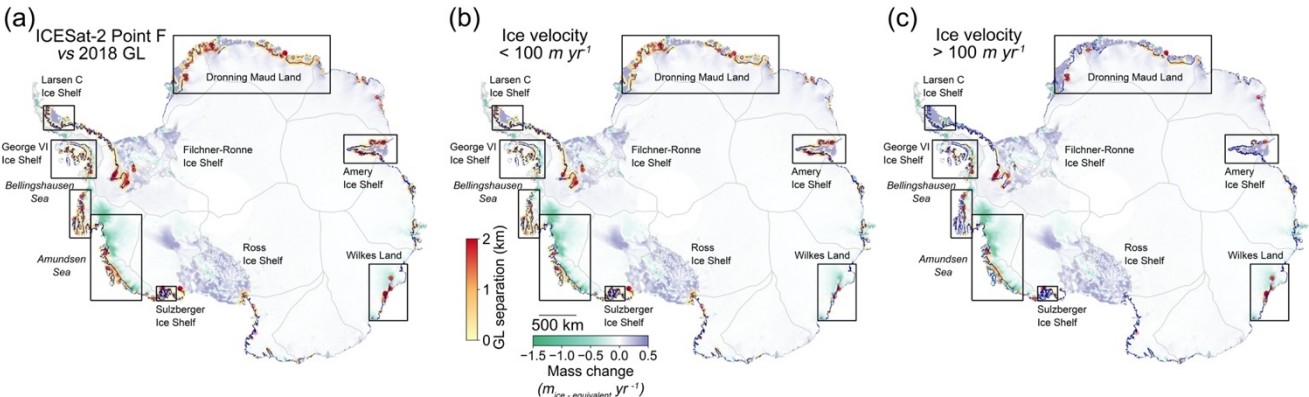

**Figure 7: a) Absolute separations between the ICESat-2-derived landward limit of tidal flexure (Point F) and Sentinel-1a/b DInSAR-derived Point F in 2018 (Mohajerani et al., 2021). b) Absolute separations in areas where the ice velocity is lower than 100 m yr⁻¹(Rignot et al., 2017); c) Absolute separations in areas where the ice velocity is higher than 100 m yr⁻¹(Rignot et al., 2017).**




**Figure 8:** Comparison between the inland limit of tidal flexure (Point F) from repeat-track analysis for two tracks located in same region on the Dronning Maud Land under different ocean tidal amplitude ranges. (a-e) ICESat-2 repeat track analysis for two right beams from repeat cycles 6, 7 and 8 in beam pair 2 of track 145. (f-j) ICESat-2 repeat track analysis for two left beams from repeat cycles 3, 7 and 8 in beam pair 3 of track 153.



### 3.3.2 Break-in-slope Point $I_b$

We compared the ICESat-2-derived Point $I_b$ with the break-in-slope from the ASAID product (Fig. 9 and Table 3), which

was delineated from Landsat-7 optical images obtained during 1999 and 2003 based on image brightness (Bindschadler et al., 2011). The positional accuracies of ASAID Point $I_b$ range from ±52 m for land and ocean terminating to ±502 m for outlet glaciers (Bindschadler et al., 2011). The mean absolute separation and standard deviation for the whole Antarctica Ice Sheet between ICESat-2-derived Point $I_b$ and ASAID Point $I_b$ are 0.43 km and 0.43 km, respectively (Table 3). On Larsen C Ice Shelf, the mean absolute separation and standard deviation are lowest, which are 0.19 km and 0.17 km, respectively.

Larsen C Ice Shelf in general is a slow-moving mountainous region and the ASAID Point $I_b$ is a good representation of the grounding line (Li et al., 2020). In similar regions with slow ice flow and steep surface gradients such as Sulzberger Ice Shelf and George VI Ice Shelf, the GL separations are also small (Table 3). The highest separations are located in the fast flowing regions of Amundsen Sea Embayment, the mean absolute separation and standard deviation are 1.42 km and 1.23 km, respectively. For comparison, we also calculated the separations between the ICESat-2-derived Point $I_b$ and the

DInSAR-derived Point F in 2018 (Fig. 10 and Table 4). Although Point F and Point $I_b$ are two different grounding zone features derived from different techniques, in regions apart from ice plain, these two features should be close in locations (Brunt et al., 2011; Christie et al., 2016). The mean absolute separation and standard deviation between ICESat-2-derived Point $I_b$ and DInSAR-derived Point F in 2018 over Antarctica Ice Sheet are 0.02 km and 0.02 km (Table 4), respectively, which are of the same magnitudes with the ICESat-2-derived Point F. Over the fast flowing ice streams of Amundsen Sea

Embayment, the mean absolute separation and standard deviation are 0.04 km and 0.04 km, respectively, much lower than the ASAID Point $I_b$.

Detailed spatial distribution maps of the ICESat-2-derived Point $I_b$, as well as four other grounding zone products, including the DInSAR-derived Point F in 2017 at Thwaites Glacier (Milillo et al., 2019), the DInSAR-derived Point F in 2018 (Mohajerani et al., 2021), the ASAID Point $I_b$ during 1999 and 2003 (Bindschadler et al., 2011), as well as the ICESat-

derived Point $I_b$ (Brunt et al., 2010a), are shown in Fig. 11. In mountainous region with stable grounding line, such as the Sulzberger Ice Shelf (Fig. 11d), different grounding zone products match well with each other. On fast flowing glaciers where the subglacial bed and surface slopes are shallow, Point $I_b$ is difficult to identify from satellite imagery based on a change in image brightness (Bindschadler et al., 2011; Christie et al., 2016, 2018). This reflects on the Pope and Smith Glaciers (Fig. 11g), where Point $I_b$ from ASAID product cannot identify the correct ice sheet boundary. In addition, on the

fast-flowing Jutulstraumen Glacier (ice velocity > 700 m yr$^{-1}$) at Dronning Maud Land (Fig. 11b), there exists a up-to 15 km deviation between the ICESat-2-derived Point $I_b$ and ASAID Point $I_b$. On the contrary, in regions which have been experiencing ice dynamical thinning and rapid grounding line retreats along the Amundsen Sea Embayment (Chuter et al., 2017; Smith et al., 2020; Bamber and Dawson, 2020), the ICESat-2-derived Point $I_b$ have good agreement with the latest DInSAR-derived Point F in 2018 (Figs. 11e-i). Moreover, they both show a pervasive retreat compared with the ASAID

Point $I_b$ identified between 1999 and 2003, especially on the fast-flowing glaciers of sustained grounding line retreat, such as



the Berry Glacier in Fig. 11e, the unnamed glaciers along Getz Ice Shelf in Fig. 11f, the Pope, Smith and Kohler Glaciers in Fig. 11g, the Thwaites Glacier in Fig. 11h and the Pine Island Glacier in Fig. 11i. In the 'butterfly' region of the Thwaites Glacier, which is featured by rapid ice thinning and grounding line retreat during the past two decades (Milillo et al., 2019), there is an almost 10 km landward migration between the ASIAD Point $I_b$ and the ICESat-2-derived Point $I_b$ (Fig. 3g).

Similar to Thwaites Glacier, Getz Ice Shelf has also been experiencing ice dynamical thinning (Selley et al., 2021) and grounding line retreat (Christie et al., 2018; Konrad et al., 2018). The ICESat-2-derived Point $I_b$ at an unnamed fast flowing glacier shows an approximately 6 km landward migration compared with ASAID break-in-slope (Fig. 3m).

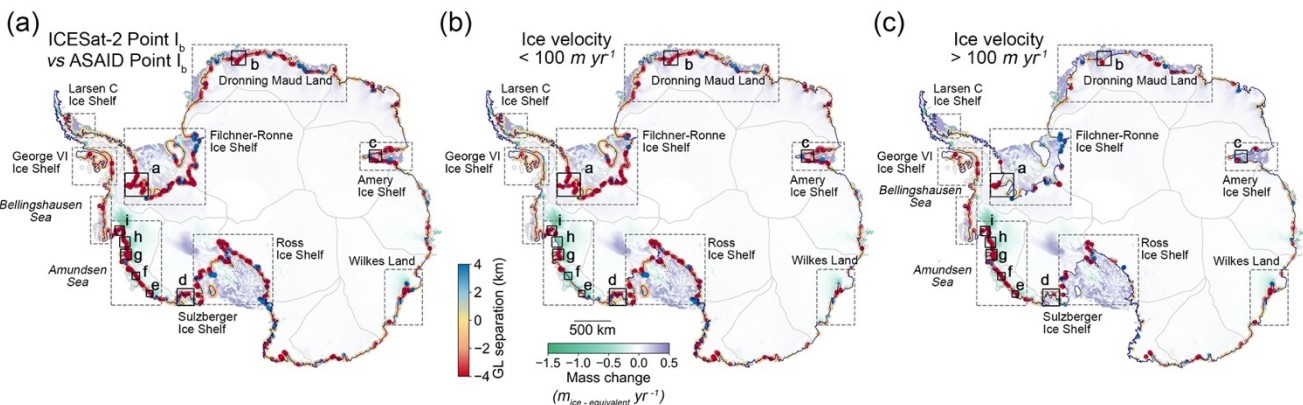

**Figure 9: a) Separations between the ASAID-derived break-in-slope and ICESat-2-derived break-in-slope (negative value is**
**retreating while positive value is advancing). b) Regions with ice velocity (Rignot et al., 2017) less than 100 m yr⁻¹; c) Regions with**
**ice velocity larger than 100 m yr⁻¹. The black boxes denote the spatial extents of regions mapped in Figure 11.**

**Table 3: Mean absolute separation (km) and standard deviation (km) between ICESat-2-derived break-in-slope (Point $I_b$) and ASAID break-in-slope product (Bindschadler et al., 2011) in individual regions.**

| Region | Ice velocity < 100 m/yr | | | Ice velocity > 100 m/yr | | | All | |
|---|---|---|---|---|---|---|---|---|
| | Mean Absolute separation (km) | Standard deviation (km) | Ratio | Mean Absolute separation (km) | Standard deviation (km) | Ratio | Mean Absolute separation (km) | Standard deviation (km) |
| Antarctica | 0.36 | 0.36 | 0.71 | 0.59 | 0.54 | 0.27 | 0.43 | 0.43 |
| Ross Ice Shelf | 0.42 | 0.41 | 0.77 | 0.62 | 0.58 | 0.2 | 0.44 | 0.43 |
| Filchner-Ronne Ice Shelf | 0.46 | 0.49 | 0.76 | 0.55 | 0.49 | 0.22 | 0.48 | 0.49 |
| Larsen C Ice Shelf | 0.19 | 0.17 | 0.94 | 0.25 | 0.19 | 0.06 | 0.19 | 0.17 |
| Dronning Maud Land | 0.59 | 0.59 | 0.77 | 0.72 | 0.68 | 0.22 | 0.6 | 0.59 |
| Amery Ice Shelf | 0.75 | 0.77 | 0.86 | 0.62 | 0.55 | 0.13 | 0.73 | 0.74 |
| Amundsen Sea | 1.23 | 1.24 | 0.33 | 1.42 | 1.23 | 0.67 | 1.33 | 1.22 |
| Bellingshausen Sea | 0.42 | 0.37 | 0.56 | 0.62 | 0.55 | 0.43 | 0.48 | 0.41 |
| Wilkes Land | 0.34 | 0.31 | 0.24 | 0.49 | 0.44 | 0.73 | 0.46 | 0.42 |
| Sulzberger Ice Shelf | 0.18 | 0.14 | 0.88 | 0.4 | 0.3 | 0.12 | 0.23 | 0.2 |
| George VI Ice Shelf | 0.2 | 0.18 | 0.63 | 0.52 | 0.51 | 0.36 | 0.26 | 0.24 |






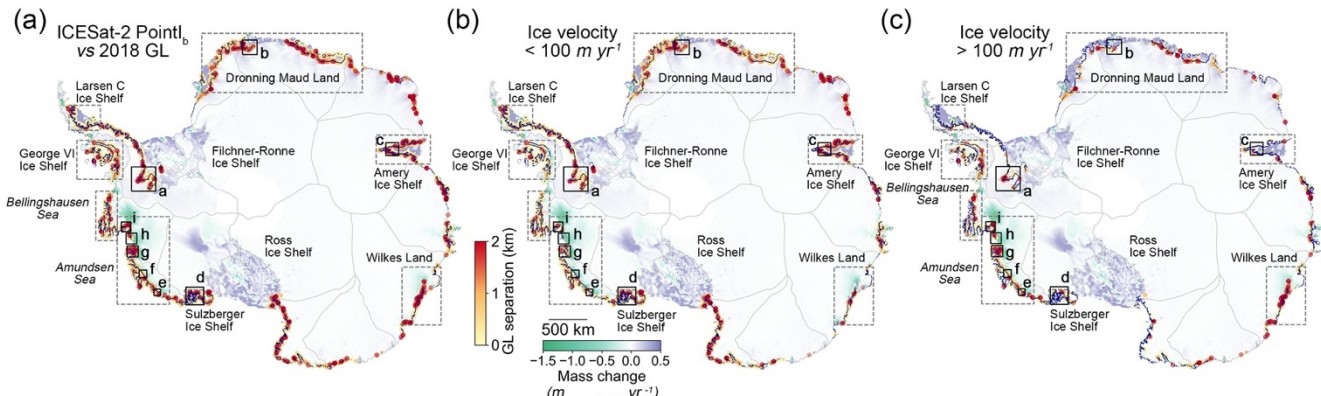

**Figure 10: a) Absolute separations between ICESat-2-derived break-in-slope (Point I$_b$) and 2018 DInSAR-derived Point F (Mohajerani et al., 2021). b) Regions with ice velocity (Rignot et al., 2017) less than 100 m yr$^{-1}$; c) Regions with ice velocity larger than 100 m yr$^{-1}$. The black boxes denote the spatial extents of regions mapped in Figure 11.**


**Table 4: Mean absolute separation (km) and standard deviation (km) between ICESat-2-derived break-in-slope (Point I$_b$) and 2018 DInSAR-derived Point F (Mohajerani et al., 2021) in individual regions**

| Region | Ice velocity < 100 m/yr | | | Ice velocity > 100 m/yr | | | All | |
| --- | --- | --- | --- | --- | --- | --- | --- | --- |
| | Mean Absolute separation (km) | Standard deviation (km) | Ratio | Mean Absolute separation (km) | Standard deviation (km) | Ratio | Mean Absolute separation (km) | Standard deviation (km) |
| Antarctica | 0.01 | 0.01 | 0.78 | 0.08 | 0.09 | 0.22 | 0.02 | 0.02 |
| Larsen C Ice Shelf | 0.02 | 0.02 | 0.95 | 0.26 | 0.32 | 0.05 | 0.02 | 0.02 |
| Dronning Maud Land | 0.13 | 0.15 | 0.81 | 0.18 | 0.22 | 0.19 | 0.13 | 0.15 |
| Amery Ice Shelf | 0.2 | 0.28 | 0.91 | 0.11 | 0.14 | 0.09 | 0.03 | 0.04 |
| Amundsen Sea | 0.01 | 0.01 | 0.51 | 0.04 | 0.04 | 0.49 | 0.01 | 0.01 |
| Bellingshausen Sea | 0.02 | 0.02 | 0.81 | 0.07 | 0.08 | 0.19 | 0.02 | 0.02 |
| Wilkes Land | 0.16 | 0.18 | 0.36 | 0.83 | 0.95 | 0.64 | 0.42 | 0.48 |
| Sulzberger Ice Shelf | 0.01 | 0.01 | 0.93 | 0.15 | 0.25 | 0.07 | 0.01 | 0.01 |
| George VI Ice Shelf | 0.01 | 0.01 | 0.71 | 0.11 | 0.14 | 0.29 | 0.01 | 0.01 |



**Figure 11:** Spatial distributions of ICESat-2-derived break-in-slope (Point I$_b$) in each individual region (black boxes in Figs. 9 and 10). For comparison, ICESat-derived Point I$_b$ locations are shown as the yellow dots (Brunt et al., 2010a); ASAID-derived Point I$_b$ is shown as the green line (Bindschadler et al., 2011); DInSAR-derived Point F in 2018 is shown as the pink line (Mohajerani et al., 2021); DInSAR-derived Point F in 2017 on Thwaites Glacier (panel h) is shown as the blue line (Milillo et al., 2019). In all subplots, data are superimposed over recent ice surface velocity magnitudes (Rignot et al., 2017) in Antarctica polar stereographic projection (epsg:3031).

### 3.3.3 Inshore limit of hydrostatic equilibrium - Point H

The inshore limit of hydrostatic equilibrium Point H mapped from the ASAID project is the most complete product for Point H to date, which was derived from ICESat-derived Point H and the Landsat-7 imagery (Bindschadler et al., 2011). The positional error of Point H from ASAID product is about 2 km. The absolute separation between the ICESat-2-derived Point H and ASAID Point H are shown in Fig. 12. The overall mean absolute separation and standard deviation for the whole



Antarctica Ice Sheet between the two products are 1.65 km and 1.29 km (Table 5), respectively, which are within the 2 km geolocation error of ASAID Point H. However, they vary with different regions (Fig. 12 and Table 5). The Larsen C Ice Shelf has the smallest mean absolute separation and standard deviation, while the Amery Ice Shelf has the highest mean

absolute separation and standard deviation of 2 km and 1.62 km, respectively.

The location of Point H is not stagnant but changes with ocean tides. On the western flank of the Skytrain Ice Rise on the Filchner-Ronne Ice Shelf (Fig. 13), the ICESat-2-derived Points F along the left (Figs. 13a-e) and right beams (Figs. 13f-j) of track 1071 are separated by 158 m. However, the distance between the ICESat-2-derived Points H is 6 km. The tidal range at the seaward Point H along the left beam of track 1071 is 3.3 m while it is only 0.8 m at the landward Point H along the right

beam of track 1071. This indicates that the ocean tide oscillation will not only influence the grounding point of the ice but will also change the point of hydrostatic equilibrium. More examples will be used to fully investigate the influence of ocean tides on the grounding zone width in future research.

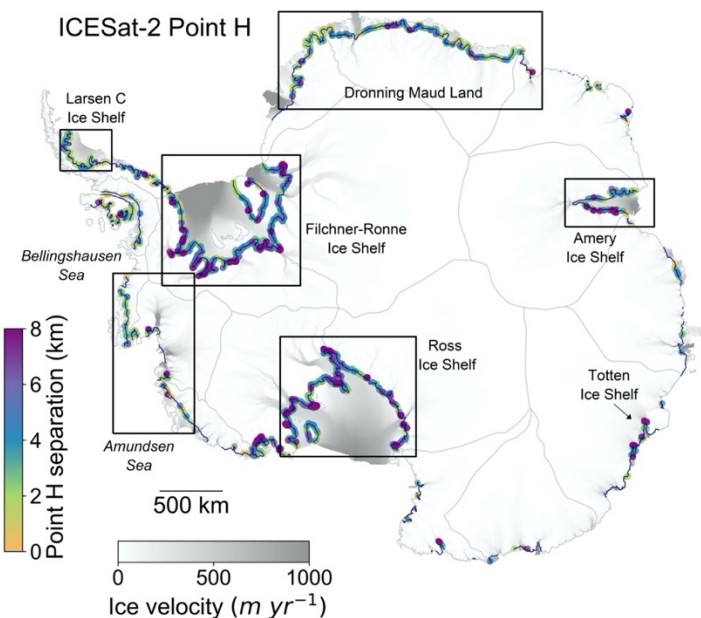

**Figure 12: The absolute separations between the ICESat-2-derived Point H and the Point H from ASAID grounding line project**
**(Bindschadler et al., 2011). Data are superimposed over recent ice surface velocity magnitudes (Rignot et al., 2017) in Antarctic polar stereographic (epsg:3031) projection.**

**Table 5: The mean absolute separation and standard deviations between ICESat-2-derived Point H and ASAID-derived Point H (Bindschadler et al., 2011).**

| Region | Mean absolute separation (km) | Standard deviation (km) |
|---|---|---|
| Antarctica | 1.65 | 1.29 |
| Ross Ice Shelf | 1.66 | 1.30 |
| Filchner-Ronne Ice Shelf | 1.70 | 1.33 |
| Larsen C Ice Shelf | 1.35 | 0.90 |



| Dronning Maud Land | 1.42 | 1.09 |
| Amery Ice Shelf | 2.0 | 1.62 |
| Amundsen-Bellingshausen Seas | 1.34 | 0.91 |

**Figure 13: Comparison between the inshore limit of hydrostatic equilibrium (Point H) from repeat-track analysis for left and right beams of track 1071 located in the Filchner-Ronne Ice Shelf under different ocean tidal amplitude ranges. (a-e) ICESat-2 repeat track analysis for two left beams from repeat cycles 3, 5 and 7 in beam pair 3 of track 1071. (f-j) ICESat-2 repeat track analysis for two right beams from repeat cycles 5, 6 and 7 in beam pair 3 of track 1071.**





## 4 Discussion


Although good agreement exists between the ICESat-2-derived Point F and DInSAR-derived Point F in 2018, large deviations have been observed in slow-moving regions due to short-term grounding line migrations over ice plain caused by ocean tides. The DInSAR-derived Point F using Sentinel 1-a/b interferograms in 2018 sampled different grounding line positions with changes in ocean tides, however, it fails to capture the ephemeral grounding observed in this study (Figs. 8a-

e). This indicates that one year worth of DInSAR data may not be fully adequate to address the migration of grounding line in different ocean tide amplitudes within a tidal cycle (Mohajerani et al., 2018).

By comparing the ICESat-2-derived Point F with ICESat-2 crossovers, as well as several published grounding zone products on Filchner-Ronne Ice Shelf and Ross Ice Shelf, we are able to detect the possible errors in different grounding zone products. The large landward deviations of ESA CCI DInSAR-derived Point F on the western flank of Support Force Glacier

in 2016 and the northern flank of Parry Peninsula in 2014, compared with all the other grounding zone products, indicate the ESA CCI DInSAR-derived Point F are likely to be in error. An up-to 15 km landward grounding line migration was identified for ICESat-2-derived Point F at Crary Ice Rise compared with previous grounding line products, which is coincident with the high mass loss in this region (Smith et al., 2020), indicating it can be a possible grounding line retreat. Further research is needed to fully understand the reason why the grounding line has been retreating in this region.

In highly crevassed and fast flowing glaciers with low tidal amplitudes (Padman et al., 2002), such as the Pine Island Glacier and Thwaites Glacier located in the Amundsen Sea Embayment, it is difficult to image both Points F and H based on the dynamic method which samples elevation changes at different tidal phases using repeat track analysis. The fast movement of glaciers, however, can cause extensive advection of ice surface features on the floating ice, such as crevasses and surface undulations (Moholdt et al., 2014; Khazendar et al., 2013). This will result in high elevation anomalies not associated with

ocean tides, making it difficult to identify the limit of ice flexure of the grounding zone. The Lagrangian framework has been used to reduce the elevation change anomalies caused by feature advection (Moholdt et al., 2014; Dutrieux et al., 2013). This method, however, requires the movement of ice features synchronized with the ice flow, which is only applicable on floating ice shelves (Marsh et al., 2016). Thus it is not suitable for this study as we are only interested in the transition between grounded ice and floating ice. Unlike the limit of tidal flexure Points F and H that directly depend on the tidal variations, the

break-in-slope point is the location where the ice 'feels' the bed sufficiently to react to the stresses associated with this contact and it is not influenced by the temporal tidal variations (Bindschadler et al., 2011). Also the elevation differences measured by Points F and H are always noisier than the absolute surface elevation measured by Point $I_b$. The static method developed in this study is able to reliably detect the break-in-slope even in highly crevassed fast flowing glaciers, such as Thwaites Glacier and Getz Ice Shelf, where the break-in-slope is less prominent and the optical imagery approaches are

normally unable to interpret the correct break-in-slope (Bindschadler et al., 2011; Rignot et al., 2011a).



Compared with the ASAID break-in-slope delineation from Landsat-7 images, large landward deviations exist both in ICESat-2-derived Point $I_b$ and the latest DInSAR-derived Point F in 2018. These landward deviations can be possibly attributed to the grounding line retreat, given the fact that dynamical mass loss has been taking place in this region (Smith et al., 2020). However, as the ICESat-2-derived Point $I_b$ and ASAID Point $I_b$ are calculated based on two different methods,
there will always be differences between these two boundaries due to data quality or incorrect interpretation (Bindschadler et al., 2011), therefore caution is needed when identifying the true grounding zone retreat. The Antarctica grounding zone product produced in this study uses only 18 months of ICESat-2 ATL06 datasets, with more repeat cycles coming in next few years, we will be able to map the grounding zone over the same region repeatedly and efficiently using the same techniques. This could reduce the errors in interpreting grounding line migrations and contribute to a more accurate
assessments of grounding line retreat and ice sheet instability.

## 5 Summary and conclusion

We presented the first ICESat-2-derived high-resolution Antarctica grounding zone product using just 18 months of data, including three grounding zone features (Li et al., 2021). This product has been derived using automated techniques developed in this study based on ICESat-2 repeat tracks, and has been validated using a crossover analysis of ICESat-2 data
over Filchner-Ronne Ice Shelf and Ross Ice Shelf and against latest DInSAR measurements over the Antarctica Ice Sheet. A total of 21346 the landward limit of ice flexure (Point F), 18149 the inshore limit of hydrostatic equilibrium (Point H), and the 36765 the break-in-slope (Point $I_b$) were identified for the Antarctica Ice Sheet. This not only represents a significant increase in grounding zone density compared with ICESat measurements, but also achieves an improved coverage. The mean absolute separation and standard deviation between the ICESat-2-derived Point F and the DInSAR-derived grounding
line product in 2018 are 0.02 km and 0.02 km, respectively, comparable to the precision of the DInSAR product. While the dynamic method can have difficulties defining the grounding zone in highly-crevassed fast flowing glaciers with low tidal range, such as the fast flowing glaciers in Amundsen Sea Embayment, the static method is able to retrieve the break-in-slope reliably in these regions. The mean absolute separation and standard deviation between the ICESat-2-derived Point $I_b$ and the DInSAR-derived Point F in 2018 over the fast-flowing regions at Amundsen Sea Embayment are 0.04 km and 0.04 km,
respectively. Additionally, both ICESat-2-derived Point $I_b$ and the DInSAR-derived Point F show pervasive landward migrations compared with the ASAID product. This coincides with the consistent mass loss and grounding line retreat in this region.

Although our study period only covers 18 months, we are able to detect the short-term grounding zone migrations due to ocean tide oscillation. Examples of repeat track analysis on the Dronning Maud Land and the Filchner-Ronne Ice shelf show
that the influence of ocean tide variations will not only change the grounding location of the ice but will also influence the point of fully hydrostatic equilibrium for the floating ice. A more detailed research on the relationship between ocean tide variations, grounding zone width and different geophysical factors is needed in future. With more repeat cycles coming out



in next few years, we will be able to map the grounding zone features based on the same techniques developed in this study repeatedly and efficiently. This will allow for tracking GL migrations at higher accuracy and provide more comprehensive insights into ice sheet instability, which is valuable for both cryosphere and sea level science communities.

## 6 Data availability

The dataset produced in this study is available at the University of Bristol data repository, data.bris, at https://doi.org/10.5523/bris.bnqqyngt89eo26qk8keckglww (Li et al., 2021). It is also archived and maintained at the National Snow and Ice Data Center (NSIDC). The ICESat-2 data used in this study are available from the NSIDC.

## Acknowledgements

TL was supported by the China Scholarship Council (CSC) – University of Bristol joint-funded PhD scholarship. JLB and SJC were supported by European Research Council grant number 694188 (GlobalMass).

## Author contribution

TL developed the methods, produced the results and wrote the paper. GJD and SJC assisted with data processing. JLB conceived the study and contributed to the interpretation of the results. All authors commented on the manuscript.

## Competing interests

The authors declare no completing interest.

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



**Appendices**

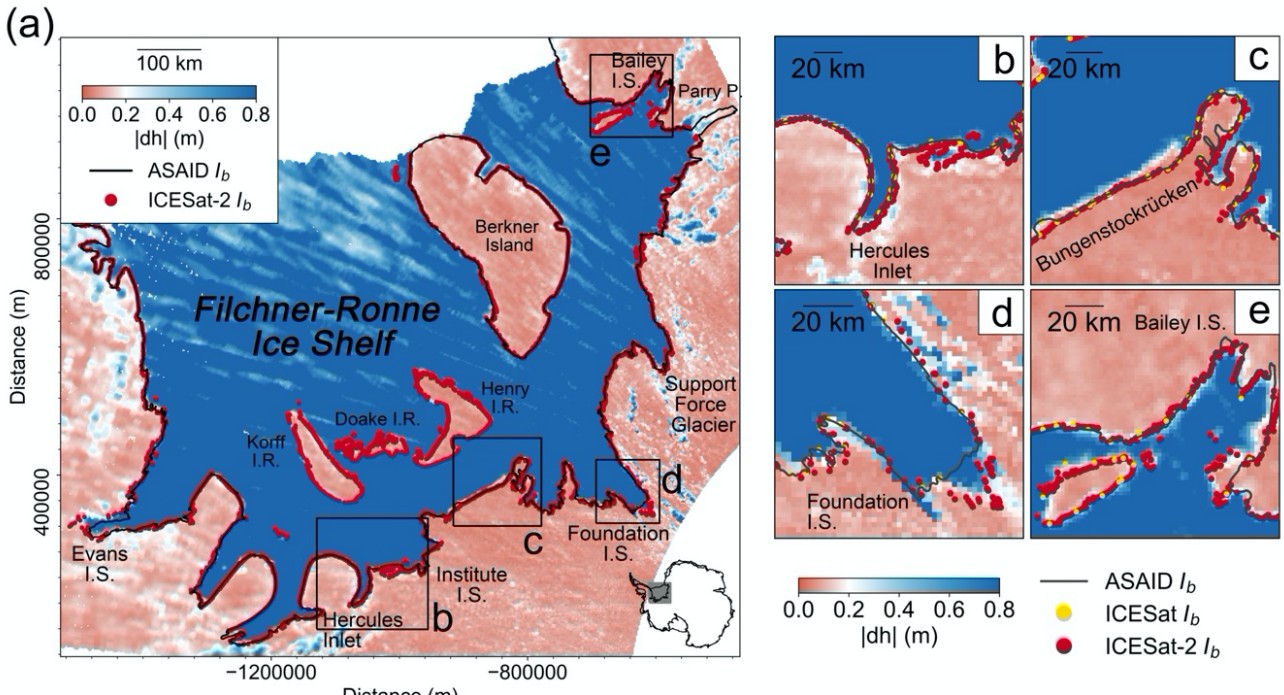

**Figure A1: (a) Spatial distribution of the absolute elevation change at ICESat-2 crossovers per 2 km grid cell across the Filchner-Ronne Ice shelf overlaid with ICESat-2-derived break-in-slope (Point $I_b$), the four black boxes denote the individual regions plotted in b-e). b) Hercules Inlet; c) Bungenstockrücken;  d) Support Force Glacier; e) Bailey Ice Stream. In all subplots, the**
**ICESat-2-derived break-in-slope (Point $I_b$) are shown as red dots. The ICESat-derived Point $I_b$ are shown as the yellow dots. The ASAID Point $I_b$ is shown as the black line (Bindschadler et al., 2011).**





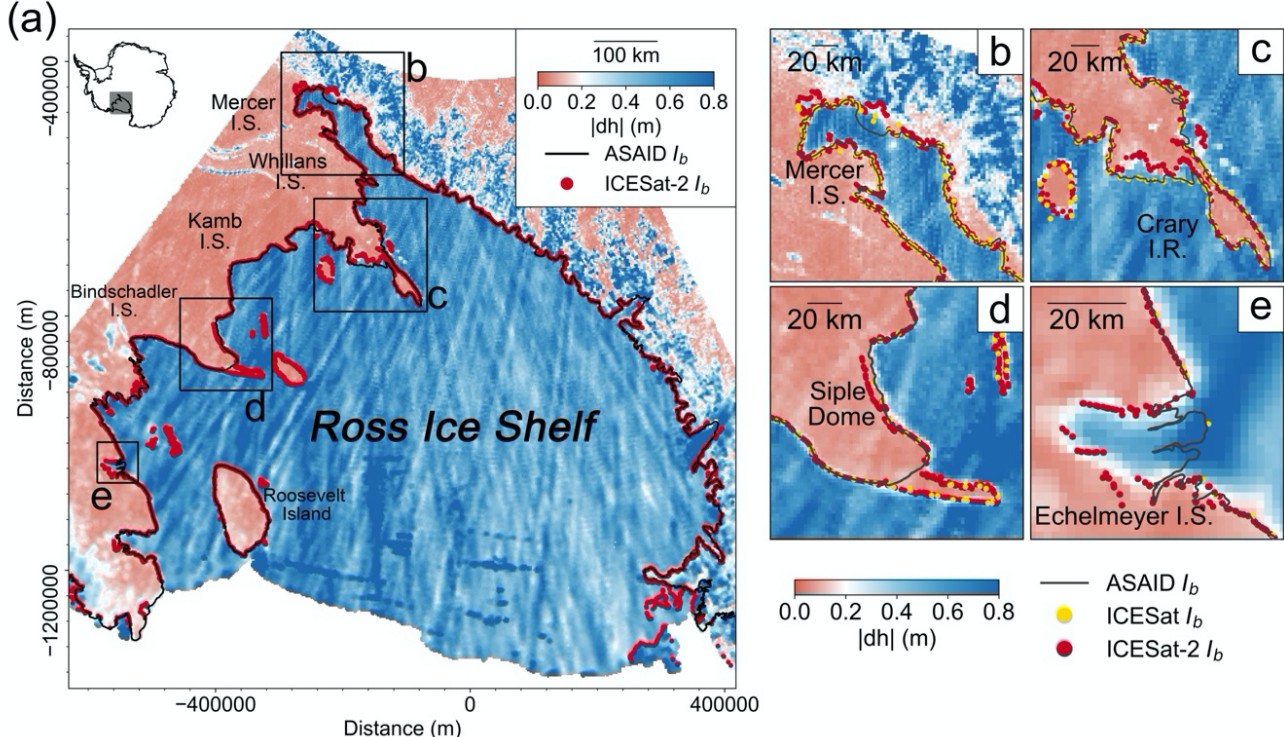

**Figure A2: (a) Spatial distribution of the absolute elevation change at ICESat-2 crossovers per 2 km grid cell across the Ross Ice shelf overlaid with ICESat-2-derived break-in-slope (Point I$_b$), the four black boxes denote the individual regions plotted in b-e). b) Mercer Ice Stream; c) Crary Ice Rise; d) Siple Dome; e) Echelmeyer Ice Stream. In all subplots, the ICESat-2-derived break-in-slope (Point I$_b$) are shown as red dots. The ICESat-derived Point I$_b$ are shown as the yellow dots. The ASAID Point I$_b$ is shown as the black line (Bindschadler et al., 2011).**

**Table A1. List of different grounding line (GL) products used to update the Depoorter et al. (2013) grounding line for the composite grounding line generated in Section. 2.2.**

| Region | Grounding line product |
| --- | --- |
| Larsen C Ice Shelf | ESA Climate Change Initiative (CCI) GL between 2015 and 2016 (ESA, 2017) |
| Recovery Glacier | ESA CCI GL in 2014 |
| Getz Ice Shelf | ESA CCI GL in 2017 |
| Pine Island Glacier | DInSAR GL in 2015 (Milillo et al., 2017) |
| Thwaites Glacier | ESA CCI GL in 2016 |
| Smith, Pope and Kohler Glaciers | MEaSUREs GL in 2011 (Rignot et al., 2016) ESA CCI GL in 2016 |
| Moscow University Ice Shelf | Manually-defined GL based on the break-in-slope from REMA DEM (Howat et al., 2019) to account for the orientation of ICESat-2 tracks |
| Kiel Glacier | ESA CCI GL in 2016 |
| Byrd Glacier | ESA CCI GL in 2011 |
| Echelmeyer Ice Stream | CryoSat-2-derived GL in 2017 (Dawson and Bamber, 2020) |