# Peer review of "A High-Resolution Antarctic Grounding Zone Product from ICESat-2 Laser Altimetry"

_Earth System Science Data, 2021_

## Referee Comment (RC2)

This manuscript presents a high-resolution antarctic grounding zone product from ICESat-2 laser altimetry. The data presented in this manuscript is critical for assessing ice sheet stability, estimating mass budget and its contribution to future sea level rise of Antarctic ice sheet, and ice sheet model projections. This study could potentially make a valuable contribution to studying Antarctic ice sheet mass balance. However, I do not believe the presentation of the manuscript at this stage is sufficiently good to warrant publication. There are some issues with the manuscript that would be valuable to address.

Major comments:
1. In Sect. 2.5, the paper describes the operation of crossover analysis, not the method of GZ features extraction. Crossover analysis is mainly to validate the results. So, Sect. 2.5 should be merged into Sect. 3.2.

2. In Sect. 3.2, the paper mainly focuses on the comparison with ICESat-2 crossover measurements. However, there are many a lot of comparisons with other study, are mixed in. These comparisons are better to be move into in Sect. 3.3.

The paper is wrote well entirely. However, there are still some problems in English writing. It is suggested to revise the English description entirely.

Other comments:

1. In Fig 2,3,8 and 13, lines of different colors are difficult to distinguish, please try changing their linetype, thickness, or color. The legend covers the data, please adjust the position.

2. Line 161 and 219, (d.I,n) should be (d, i, n).

3. In Fig 7, 9 and 10, the data source of the mass chang used in the figures is not given in the manuscript. Why use mass change? Is ice velocity more appropriate here? Moreover, these diagrams do not show absolute separations very well. I suggest showing only the key areas as shown in Fig 11

4. In Fig, suggest adding a subgraph representing the location.

---

## Author Comment (AC1)

Dear Editor and Reviewers,

Thank you for your time in reviewing our manuscript and providing detailed comments, which have proven very helpful in revising this paper. We have now revised the manuscript by fully addressing the suggested changes. Below you will find our detailed responses to each comment presented by the reviewers in blue text.

Yours sincerely,

Tian Li

**Reviewer 1**

The manuscript uses 18 months of ICESat-2 repeat tracks to identify the grounding zone of the Antarctica Ice Sheet. It develops a method that could automated mapping the grounding zone and produces a grounding zone product that has nearly complete coverage of the Antarctica Ice Sheet. Elevation changes derived from ICESat-2 ascending and descending cross-over passes are also used for validation.

Given the significant increase in grounding zone density and the improved coverage, I believe that this ICESat-2-derived Antarctica grounding zone product will be of large interest to the cryosphere community. Furthermore, with more ICESat-2 repeat cycles coming out in next few years, the dynamic changes of grounding zone could be evaluated repeatedly and efficiently based on the automated techniques developed in this study. The manuscript is overall well written. I have no major comments except a few suggestions listed below. Therefore, I look forward to seeing this paper published in Earth System Science Data.

***Specific comments:***

Line 27: There is no 'Point G' in Fig. 1.

Agree and corrected. We have now added a new figure showing the grounding zone structure as Figure 1 which now has 'Point G'.

Line 37: Suggest adding a figure to show the schematic of grounding zone. This will clearly show where the point F, H, Im and Ib located.

Thank you for the suggestion. We agree that a figure of grounding zone structure will clearly show the locations of different grounding zone features, we have now added a new figure (Figure 1 below) to show the schematic of grounding zone.

[Figure]

Figure 1. Schematic diagram of the ice shelf grounding zone structure adapted from Fricker and Padman (2006). Point G is the true grounding line where the grounded ice first gets in contact with the ocean, Point F is the

landward limit of ice flexure caused by ocean tidal movement, Point H is the seaward limit of ice flexure and the inshore limit of hydrostatic equilibrium, Point $I_b$ is the break in surface slope, Point $I_m$ is the elevation minimum inside the grounding zone.

Figure 2, 8 &13: The legend overlaps with the curve. Considering change the position of the legend for clarity.

Thank you for the suggestion. We have now changed the positions of all the previously overlapped legends in Figures 2, 8 and 13 (now Figures 3, 9 and 14).

Section 3.3.2: Hogg et al. (2018) also detected the Antarctica Ice Sheet break-in-slope point $I_b$ using another altimetry data (CryoSat-2). I suggest the ICESat-2-derived Point $I_b$ could also be compared to that product.

Reference: Hogg, A. E., A. Shepherd, L. Gilbert, A. Muir, and M. R. Drinkwater (2018), Mapping ice sheet grounding lines with CryoSat-2, Advances in Space Research, 62(6), 1191-1202.

Thank you for the comment. Unfortunately the Hogg et al. (2018) dataset is not publicly available online and as their paper focused on describing a new method developed for mapping the break-in-slope from CryoSat-2 observations, rather than presenting the break-in-slope dataset itself, we could not include the comparison. In addition, their method has not been applied to the whole Antarctic Ice Sheet but only tested on four regions: Filchner-Ronne Ice Shelf, Ekstrom Ice Shelf, Larsen C Ice Shelf and Amundsen Sea sector. By comparing our ICESat-2-derived Point $I_b$ with three grounding line mapping results across the Antarctic continent, including the ASAID break-in-slope, Sentinel-1a/b DInSAR-derived Point F and the ICESat-2 crossover measurements, we are confident that the overall quality of ICESat-2-derived Point $I_b$ and the grounding line changes are fully assessed in our study. Therefore, comparing our results with the CryoSat-2-derived Point $I_b$ in Hogg et al. (2018), if it was available, would have provided little or no added value to the current study. In fact, this would start to become more of a comparison of different approaches rather than a validation of the approach we present in the paper, which is not appropriate for this journal or our publication although it could be interesting to do in different context.

**Reviewer 2**

This manuscript presents a high-resolution antarctic grounding zone product from ICESat-2 laser altimetry. The data presented in this manuscript is critical for assessing ice sheet stability, estimating mass budget and its contribution to future sea level rise of Antarctic ice sheet, and ice sheet model projections. This study could potentially make a valuable contribution to studying Antarctic ice sheet mass balance. However, I do not believe the presentation of the manuscript at this stage is sufficiently good to warrant publication. There are some issues with the manuscript that would be valuable to address.

We thank the reviewer for the detailed comments. We have now made significant changes to the presentation of the manuscript in response to the comments. We have improved the figures by changing the legend locations and line styles. We have also incorporated the comparisons between ICESat-2-derived grounding zone features with other studies described in the Section 3.2 "Comparison with ICESat-2 crossover measurements" into the corresponding sections focusing on each individual grounding zone feature.

***Major comments:***

1. In Sect. 2.5, the paper describes the operation of crossover analysis, not the method of GZ features extraction. Crossover analysis is mainly to validate the results. So, Sect. 2.5 should be merged into Sect. 3.2.

Thank you for the suggestion. We agree that the crossover analysis is mainly for validating the GZ features Point F and Point $I_b$ mapped from the dynamic and static methods detailed in Sections 2.3 and 2.4. Although this crossover-derived Point F is not recorded in the final grounding zone product and used only as a validation, the process of calculating this information is still part of the grounding zone feature estimation methodology. Therefore, we would like to keep the crossover analysis method description in Section 2 Data and Methodology.

2. In Sect. 3.2, the paper mainly focuses on the comparison with ICESat-2 crossover measurements. However, there are many a lot of comparisons with other study, are mixed in. These comparisons are better to be move into in Sect. 3.3.

Thank you for the comment. We have moved the comparisons between ICESat-2-derived grounding zone features with other studies into relevant sections focusing on the validation of each individual grounding zone feature. The original Section 3.2 was also split into two sub-sections Section 3.2.1 (comparison between ICESat-2 derived Point F and crossover measurements) and Section 3.3.1 (comparison between ICESat-2 derived Point $I_b$ and crossover measurements).

The paper is wrote well entirely. However, there are still some problems in English writing. It is suggested to revise the English description entirely.

Thank you for the suggestion. We have carefully revised the English throughout the manuscript and we believed the standard of English has now been greatly improved.

***Other comments:***

1. In Fig 2,3,8 and 13, lines of different colors are difficult to distinguish, please try changing their linetype, thickness, or color. The legend covers the data, please adjust the position.

Agree and amended. We have improved the figures by adjusting the line styles, line thickness and the positions of the previously overlapped legends.

2. Line 161 and 219, (d.I,n) should be (d, i, n).

Agree and amended.

3. In Fig 7, 9 and 10, the data source of the mass chang used in the figures is not given in the manuscript. Why use mass change? Is ice velocity more appropriate here? Moreover, these diagrams do not show absolute separations very well. I suggest showing only the key areas as shown in Fig 11

Thank you for the comment. The data source of the mass change is from Smith et al. (2020), we have now added this citation in all the relevant figures. We believe the use of mass change here is more appropriate than the ice velocity, because the grounding line change has been proven to be highly related to the mass imbalance of the Antarctic Ice Sheet (Bamber and Dawson, 2020; Konrad et al., 2018; Milillo et al., 2019). We agree that the figures may not show the details of each individual region, but we think it is still valuable to show the overall distribution of the grounding zone separations between different products at the continental scale, therefore we would like to keep these figures in the main manuscript. However, we have now added three new figures in the appendix (Figs. A3, A4 and A5) to show the key regions in a similar format as Fig. 11 (now Fig. 12) as suggested by the reviewer, this would allow readers to check the GZ separations in detail.

4. In Fig, suggest adding a subgraph representing the location.

Agree and amended. We have now added a sub-figure in each panel to represent the location of each region in the Antarctic Ice Sheet, please see Fig. 12.

**References cited in authors' response:**

Bamber, J. L. and Dawson, G. J.: Complex evolving patterns of mass loss from Antarctica's largest glacier, Nat. Geosci., doi:10.1038/s41561-019-0527-z, 2020.

Fricker, H. A. and Padman, L.: Ice shelf grounding zone structure from ICESat laser altimetry, Geophys. Res. Lett., doi:10.1029/2006GL026907, 2006.

Hogg, A. E., Shepherd, A., Gilbert, L., Muir, A. and Drinkwater, M. R.: Mapping ice sheet grounding lines with CryoSat-2, Adv. Sp. Res., 62(6), 1191–1202, doi:10.1016/j.asr.2017.03.008, 2018.

Konrad, H., Shepherd, A., Gilbert, L., Hogg, A. E., McMillan, M., Muir, A. and Slater, T.: Net retreat of Antarctic glacier grounding lines, Nat. Geosci., 11(4), 258–262, doi:10.1038/s41561-018-0082-z, 2018.

Milillo, P., Rignot, E., Rizzoli, P., Scheuchl, B., Mouginot, J., Bueso-Bello, J. and Prats-Iraola, P.: Heterogeneous retreat and ice melt of thwaites glacier, West Antarctica, Sci. Adv., 5(1), eaau3433, doi:10.1126/sciadv.aau3433, 2019.

Smith, B., Fricker, H. A., Gardner, A. S., Medley, B., Nilsson, J., Paolo Nicholas Holschuh, F. S., Adusumilli, S., Brunt, K., Csatho, B., Harbeck, K., Markus, T., Neumann, T., Siegfried, M. R. and Jay Zwally, H.: Pervasive ice sheet mass loss reflects competing ocean and atmosphere processes, Science (80-. )., 368(6496), 1239–1242, doi:10.1126/science.aaz5845, 2020.

---

## Author Response (AR2)

Dear Editor Dr. Tao Che,

Thank you very much for your comment. According to the peer-review comments, Reviewer 1 (https://essd.copernicus.org/preprints/essd-2021-255/#discussion) suggested that adding a schematic of grounding zone will show the grounding zone structure more clearly, from a point of view as a reader. Therefore, after serious consideration and discussion with the team, we decided to add back this figure in the introduction as suggested by the reviewer. We also feel that this would be a good reference point to the readers to better understand the presented methodology and results in this paper. Therefore, we have included the figure as requested by the reviewer.

Yours Sincerely,
Tian Li and co-authors